# Development of an AI-Based Skin Cancer Recognition Model and Its Application in Enabling Patients to Self-Triage Their Lesions with Smartphone Pictures

Aline Lissa Okita [1,*], Raquel Machado de Sousa [1], Eddy Jens Rivero-Zavala [1], Karina Lumy Okita [1], Luisa Juliatto Molina Tinoco [1], Luis Eduardo Pedigoni Bulisani [2] and Andre Pires dos Santos [1]

1   Centro de Pesquisa em Imagem, Hospital Israelita Albert Einstein, Av. Albert Einstein, 627-Jardim Leonor, São Paulo 05652-900, SP, Brazil; rachel.msousanet@gmail.com (R.M.d.S.); eddy.zvala@einstein.br (E.J.R.-Z.); karinalumy@gmail.com (K.L.O.); luisajuliatto@gmail.com (L.J.M.T.); andre.dsantos@einstein.br (A.P.d.S.)
2   Faculdade de Medicina de Jundiaí, Rua Francisco Telles, 250, Vila Arens, Jundiaí 13202-550, SP, Brazil; luis.bulisani@foxconn.com
*   Correspondence: aline.okita@alumni.usp.br; Tel.: +55-11-98676-7350

**Abstract:** Artificial intelligence (AI) based on convolutional neural networks (CNNs) has recently made great advances in dermatology with respect to the classification and malignancy prediction of skin diseases. In this article, we demonstrate how we have used a similar technique to build a mobile application to classify skin diseases captured by patients with their personal smartphone cameras. We used a CNN classifier to distinguish four subtypes of dermatological diseases the patients might have ("pigmentation changes and superficial infections", "inflammatory diseases and eczemas", "benign tumors, cysts, scars and callous", and "suspected lesions") and their severity in terms of morbidity and mortality risks, as well as the kind of medical consultation the patient should seek. The dataset used in this research was collected by the Department of Telemedicine of Albert Einstein Hospital in Sao Paulo and consisted of 146.277 skin images. In this paper, we show that our CNN models with an overall average classification accuracy of 79% and a sensibility of above 80% implemented in personal smartphones have the potential to lower the frequency of skin diseases and serve as an advanced tracking tool for a patient's skin-lesion history.

**Keywords:** artificial intelligence; dermatology; machine learning; deep learning

## 1. Introduction

The average waiting time for a consultation with a dermatologist in Brazil is around 108 days [1]. Out of 5.565 Brazilian cities, only 504 have a dermatologist; that corresponds to merely 9.1%. However, dermatology is the second most referred specialty from national primary health care. Besides waiting, patients also need to travel long distances to reach specialized centers. Such obstacles result in negative impacts on the patients, such as delayed diagnosis, higher mortality, or higher costs during investigation and treatments. Moreover, there exist indirect costs, such as function loss in body members due to more invasive surgeries or complications from treatment, and indirect consequences, such as absenteeism at work [2].

Melanoma, the most lethal skin cancer, represents less than 3% of all skin cancers in Brazil. However, the survival of patients is directly related to the time of diagnosis [3,4]. When melanoma is discovered in the early stage, within up to 5 years, a simple resection surgery increases the patient's survival rate to about 98.4%. On the other hand, melanoma discovered at an advanced stage (or metastasis) gives a patient (only) a 22.5% chance of survival [5]. In the treatment of advanced melanoma, according to clinical studies, immunotherapy is the preferred first-line therapy due to its ability to increase overall survival by 8 to 23 months [6–8]. However, this kind of therapy, which usually involves

tomography, sentinel lymph node biopsy, and other medical procedures, is much more expensive when compared with resection surgery, as mentioned earlier.

Between 2000 and 2007, in the São Paulo state of Brazil, 2740 cases of melanoma occurred, costing USD 6.33 million, where 95.8% of this amount refers to the treatment of advanced stages of melanoma (stages III and IV) [9]. From 2021, the actual cost is even higher because immunotherapy and targeted therapy can cost up to USD 77,000 per patient per year (those therapies are not reimbursed by the Brazilian Public Health System (SUS) and hence cannot be registered in this survey).

Since August 2020, the Ministry of Health of Brazil has included two immunotherapy medications (nivolumab and pembrolizumab) for unresectable and metastatic melanomas to be reimbursed for patients [10]. Some countries, like Australia, also reimburse for this type of treatment and know the risks of its economic impact if cases of advanced melanoma keep growing [11–13]. This is why many public and private initiatives have focused on the prevention and early diagnosis of this type of cancer [11–13].

Because of its economic burden, melanoma is the most explored dermatologic condition for which techniques such as machine learning and pattern recognition have been used. Several studies have reported high accuracies of around 90% in classifying skin images [14–17], demonstrating the potential for these models to assist in early diagnosis. One notable initiative in this field is the international skin imaging collaboration (ISIC), which hosts an annual challenge to design AI models for skin-lesion classification [18]. However, it is important to note that these models are based on binary classification (benign versus malignant) with the images taken by a dermoscope. Dermoscopes use polarized light and a magnifying lens that helps with seeing structures of the lesion that are not visible to the naked eye. Such equipment is expensive and usually not available for all patients or GPs (general practitioners). This is why we propose to develop an algorithm to classify lesions based on ordinary images captured by smartphones without the use of dermoscopes.

Although our research was primarily focused on melanoma skin cancer, we expanded the model to include multi-class benign lesions (Table 1). Despite not being lethal, these chronic skin diseases may have a great negative impact on patients' quality of life; skin lesions can affect body aesthetic causing psychological problems, long-term hospitalization, and high economic costs for treatment [19].

**Table 1.** Most relevant ICDs corresponding to each disease cluster (complete table is available in the Supplementary Material).

| Benign Tumors, Cysts, Scars and Callous | Pigmentation Changes and Superficial Infections | Inflammatory Diseases and Eczemas | Suspected Lesions |
|---|---|---|---|
| D18 | L57 | L40 | C44 |
| D21.9 | L80 | L41 | C44.3 |
| D22.9 | L81 | L43 | C44.4 |
| D23 | L99.0 | L44 | C44.5 |
| D36.1 | L83 | L50 | C44.6 |
| H02.6 | E70.3 | L42 | C44.9 |
| I78.1 | L56.8 | L52 | C80 |
| I781 | A63 | L53 | C84.0 |
| L91.0 | B00 | L74 | D04 |
| L82 | B02 | L75.2 | L57.0 |
| L98.0 | B07 | L85 | L41.2 |

**Table 1.** *Cont.*

| Benign Tumors, Cysts, Scars and Callous | Pigmentation Changes and Superficial Infections | Inflammatory Diseases and Eczemas | Suspected Lesions |
|---|---|---|---|
| Q82.5 | B08.1 | L11.0 | L41.4 |
| D17.3 | B35 | L70 | C43 |
| L72.8 | B36.0 | L71 | C43.0 |
| D225 | B37.9 | L73 | C43.1 |
| D23.9 | B85 | L81.7 | C43.2 |
| L72.0 | B86 | L88 | C43.3 |
| L72.9 | L00 | L95 | C43.4 |
| L90.5 | L01.0 | L97 | C43.5 |
| L84 | L02 | L90.0 | C43.6 |
| L90.6 | L03 | L93 | C43.7 |
| L05.0 | L04.9 | E80.1 | C43.8 |
| | L08.0 | K13.0 | C43.9 |
| | L08.1 | L66.0 | |
| | L30.3 | L10.0 | |
| | L05.9 | L12.0 | |
| | B00.1 | L92 | |
| | | L51 | |
| | | L13.0 | |
| | | L27 | |
| | | L57 | |
| | | L94 | |
| | | M33 | |
| | | L08.8 | |
| | | N48.1 | |
| | | L98 | |
| | | L55 | |
| | | I83.1 | |
| | | L20.9 | |
| | | L21.9 | |
| | | L22 | |
| | | L23.9 | |
| | | L24.9 | |
| | | L28 | |
| | | L29.9 | |
| | | L30 | |
| | | L56 | |
| | | L83.1 | |
| | | L58 | |
| | | L59.0 | |

Finally, we developed an AI-based application that allows the patients to take pictures of their own lesions and receive immediate feedback of the classification with a color guide to facilitate understanding of the severity of the lesion.

## 2. Material and Methods

This is a retrospective study, performed at the "Hospital Israelita Albert Einstein" (HIAE) in São Paulo, Brazil, during the period between January and October 2021. The collected data were used according to Brazilian general personal data protection law and the research was approved by ethics committees (CAAE 45310521.5.0000.0071).

### 2.1. Data Acquisition

The data were provided by the "Teledermato Project" at the Department of Telemedicine of HIAE [20].

The Teledermato project applies to patients on the waiting list for a dermatologist in the city of São Paulo. Those patients were treated at a health unit, and their skin lesions were photographed by a nurse or health technician using a mobile application developed specifically for this data collection. This application allowed the inclusion of more than one lesion for each patient. Moreover, at least three images were made of each lesion: a photo from a distance of 50 cm, a photo from a distance of 15 cm, and a photo with a lateral view of the lesion. In total, the dataset comprised 146,277 skin images.

The images were evaluated by thirteen board-certified teledermatologists (TDs). The members of the board would indicate the most probable diagnosis from a list of 212 diseases using ICD codes and would recommend one of three actions: (1) referral directly to biopsy, (2) referral to an in-person dermatologist, or (3) referral back to the GP for further orientations. Patients whose lesions were not easily evaluable (for example, due to poor image quality) would be sent back to an in-person dermatologist.

To ensure representativeness, we omitted ICD codes that had fewer than 3 cases falling into a single disease category, for example, bullous diseases or genodermatoses. We also removed from the dataset skin conditions that could not be evaluated through photographs, such as hair and nail disorders (Table 1).

The analysis resulted in 118 ICD codes, which we grouped into four clusters based on their common visual characteristics (please see Figure 1). These clusters are as follows:

1.  Pigmentation changes and superficial infections. This cluster comprises skin diseases that cause changes in skin color, such as darkening, whitening, or the appearance of black, brown, white, or red spots. Examples of skin diseases in which there are color changes are vitiligo, melasma, lentigo, ecchymosis, and other lesions caused by bacteria, viruses, and fungi. Treatment of these conditions can be provided by both dermatologists and GPs.
2.  Inflammatory diseases and eczemas. This cluster comprises disorders that can be acute or chronic. In the case of acute symptoms, such as fever, intense pruritus, and malaise, prompt evaluation in an emergency service may be necessary. For chronic cases, evaluation and treatment by a dermatologist are recommended, but there is no urgency. This cluster comprises skin diseases with multiple inflammatory causes, such as psoriasis, lichen planus, pityriasis rosea, and others.
3.  Benign tumors, cysts, scars, and calluses. This cluster includes seborrheic keratosis, keloids, nevi, and other benign skin conditions that do not require specialized evaluation or treatment.
4.  Suspicious lesions (lesions that are suspicious in terms of malignancy). This cluster comprises lesions that cannot be certainly diagnosed as benign, for example, carcinomas, melanomas, actinic keratosis, and atypical nevi. Urgent additional evaluation and treatment by a dermatologist are recommended in this case.

Images versus cluster – initial dataset

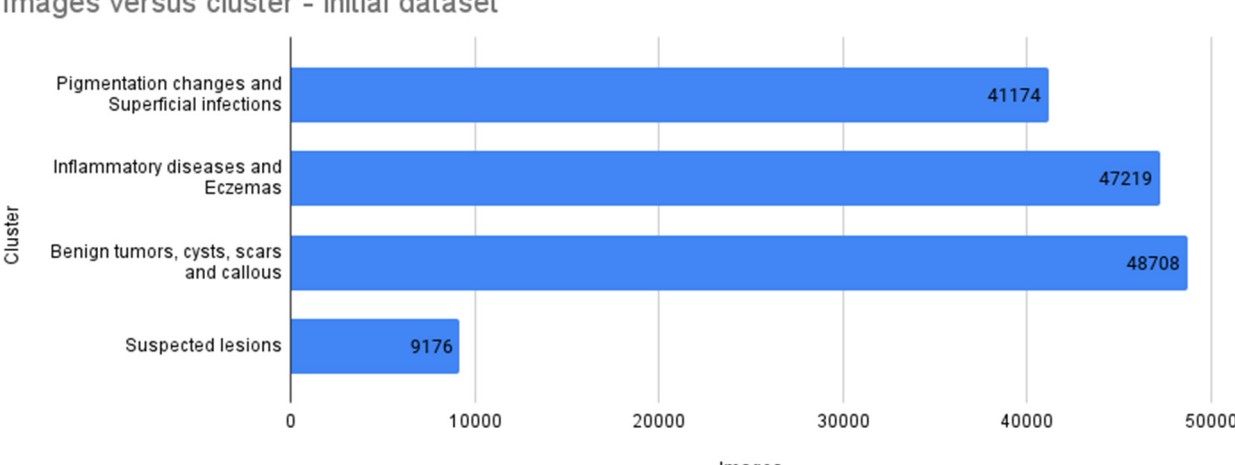

**Figure 1.** Initial dataset grouped into four clusters based on the specialist's definition.

After retrieving images corresponding to the selected ICD codes, we excluded lesions (images) without diagnosis or lesions (images) that had more than one hypothesis. This resulted in the dataset that we used for further analysis (Table 1).

As an example of the visual difference between clinical and dermoscopic images, Figure 2 shows them side by side.

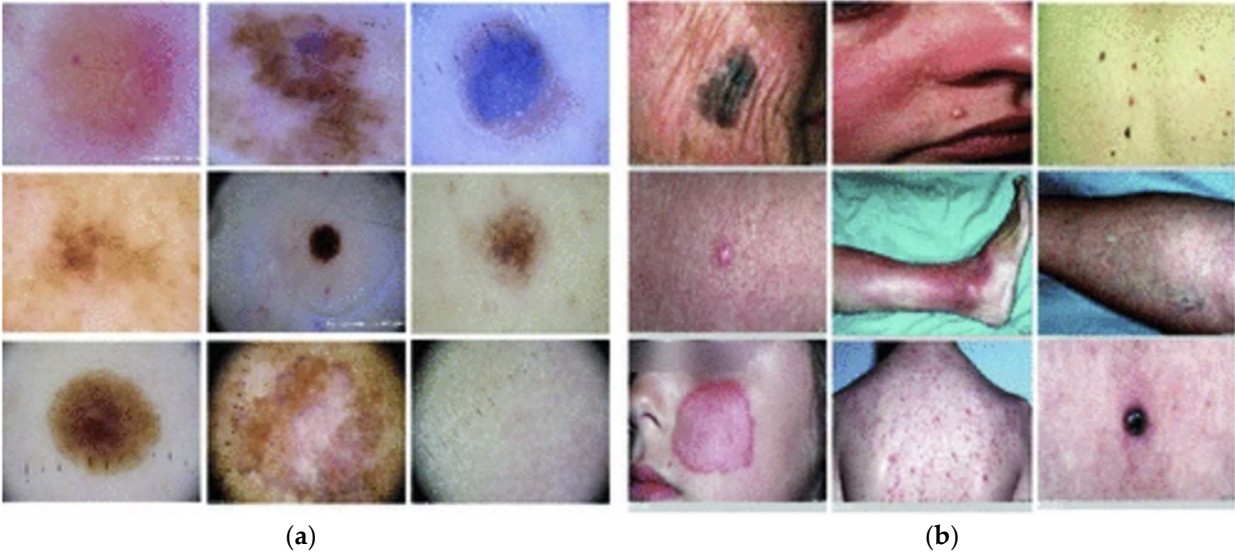

               **(a)**                                                  **(b)**

**Figure 2.** Dermoscopic and clinical image examples, shown side by side for visual comparison. Figure obtained from [21]; (**a**) Dermoscopic Images; (**b**) Clinical Images.

### 2.2. Development of the Mobile Application

The mobile application has different features. Its main structure was built to capture skin-lesion images and classify them with an AI model. The React Native library was used to develop an application that supports two types of platform (Android and iOS). The prototype of the application included features such as user registration, login, skin-lesion image history, AI image classification, guidance on further medical orientations, and a report containing the data analysis.

On the capture step, there was an image-quality evaluation related to the focus. The image focus verification was performed in the application itself using the TensorflowJS library [22]. The procedure involved calculation of the blur score, using the Laplacian filter [23] applied to the grayscale image, followed by the calculation of the variance of the

resulting image. This score was an indication of how sharp the image was. The threshold levels for this score were defined by a healthcare professional.

The AI for skin-lesion analysis was integrated with the blur detector and the interface of the mobile application. After several tests, it was discovered that this functionality was performing better on Android than on iOS, since the Teledermato dataset contained mostly images taken by Android phones.

It is worth mentioning that the application developed has the functionality of requesting acceptance of terms for use of the application.

### 2.3. Artificial Intelligence (AI) Model Development

Developing the AI model, involved the following steps: (i) selecting and preprocessing the most representative dataset; (ii) data augmentation of the training dataset; (iii) fine-tuning of the convolutional neural network (CNN) architecture of pretrained models; and (iv) an evaluation of the models.

### 2.3.1. Selection of the Dataset and Data Preprocessing

The initial dataset consisted of a total of 146,277 images. After analysis of the characteristics of the data collection process, the need for preprocessing and preparation was identified to mitigate the following problems: variation in distances between the camera and the lesion, lighting, blurred lesion images, noisy lesion images, and images with larger background areas. To minimize the variations between the distances, we used the image-similarity AI model based on the deep neural network MobileNet architecture [24] and the KNN (K—nearest neighbors) [25] technique. Those two algorithms were applied in sequence, and the resulting model was used to automatically segment the images by type of lesion. In that way, the ideal group of images was selected.

Afterwards, the dataset was further cleaned up by removing low-quality and blurry images. To detect blurred images, we used a technique based on the fast Fourier transform [26], which returned a quality score in the range between $(-\infty, \infty)$. Images with a score below "–10" were considered blurry and removed from the dataset.

The final dataset, after the cleaning procedure, consisted of 5267 images that were used for the training of the models. This dataset was divided into two parts: 80% for training and 20% for model validation and testing (resulting in 4213 for training, 527 for validation, and 527 for test). The following diagram briefly describes the dataset preprocessing.

### 2.3.2. Data Augmentation

In order to improve the generalization ability of our model and to avoid overfitting, we applied a data augmentation technique [27]. It is important to mention that this technique was applied only to the training data subset [28]. This approach ensured that the deep neural network architecture saw new variations of the data in each training iteration, including the following transformations: resizing, rotation, width and height shifting, shearing, zooming, and horizontal flipping.

### 2.3.3. Transfer-Learning and Fine-Tuned CNN Architectures of Pretrained Models

In this step, we used the transfer-learning and fine-tuning techniques applied to the convolutional neural networks. In the fine-tuning process, we froze the weights in specific layers (pretrained using ImageNet weights) and retrained the remaining ones using our data. In this way we were able to develop a model specialized in the classification of skin lesions [27,29,30].

The model was trained using both the multi-classification and binary approaches to classifying skin diseases, including suspicious lesions, benign lesions, pigmentation disorder, superficial infection, inflammatory diseases, and eczema. We selected the following deep-learning architectures to perform the training: InceptionV3 [31], InceptionResNetV2 [32], NASNetMobile [33], DenseNet201 [34], DenseNet169 [34], DenseNet121 [34], and Xception [35].

For the multi-classification training approach, a model was trained to predict 4 classes in the same classifier. We used the following training hyperparameters: softmax function [36] for classification in the final node; focal loss [37] as the model's loss function to aid performance due to class imbalance; cross-validation with early stop if the prediction error stopped decreasing after some training epochs; reduction in the learning rate if the prediction error stopped decreasing after some training iterations; and finally, the class weight as a weighting factor to fix the class-imbalance problem during training.

In the one-versus-all binary training approach, the models ran through a sequence of binary classifiers, training each of them to answer a separate ranking question. In this approach, we used the same training hyperparameters as for multi-classification except for the activation function; here, we used the sigmoid function [38] for the final classification node (Figure 3). It will be explained in the next section that the final ensemble model was developed solely with the binary training models and was chosen as our final model to integrate with the mobile application.

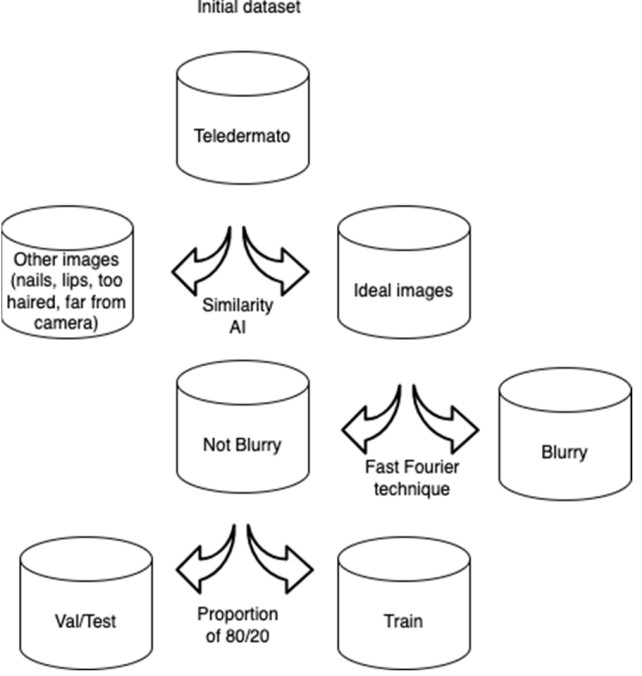

**Figure 3.** Data preprocessing steps. Starting from the Teledermato dataset, our goal was to select the relevant data, and then split them into training, validation, and test sets, to perform image classification training with deep learning.

2.3.4. Evaluation of the Models

We evaluated the models using standard metrics used by computer vision classifiers, including recall, precision, sensibility, specificity, and AUC (area under the ROC—receiver operating characteristic curve) values [39]. In addition, we utilized the Grad-Cam algorithm [40] during validation to obtain a heatmap of the image features that the model considers when making its classification, with a red color indicating more important features and a bluer color indicating less important features (Figure 4). Furthermore, we explored the possibility of using a fair gender and race validation method when predicting the validation dataset [41]. We examined the distribution of the predictions by gender and race both before and after the models were used.

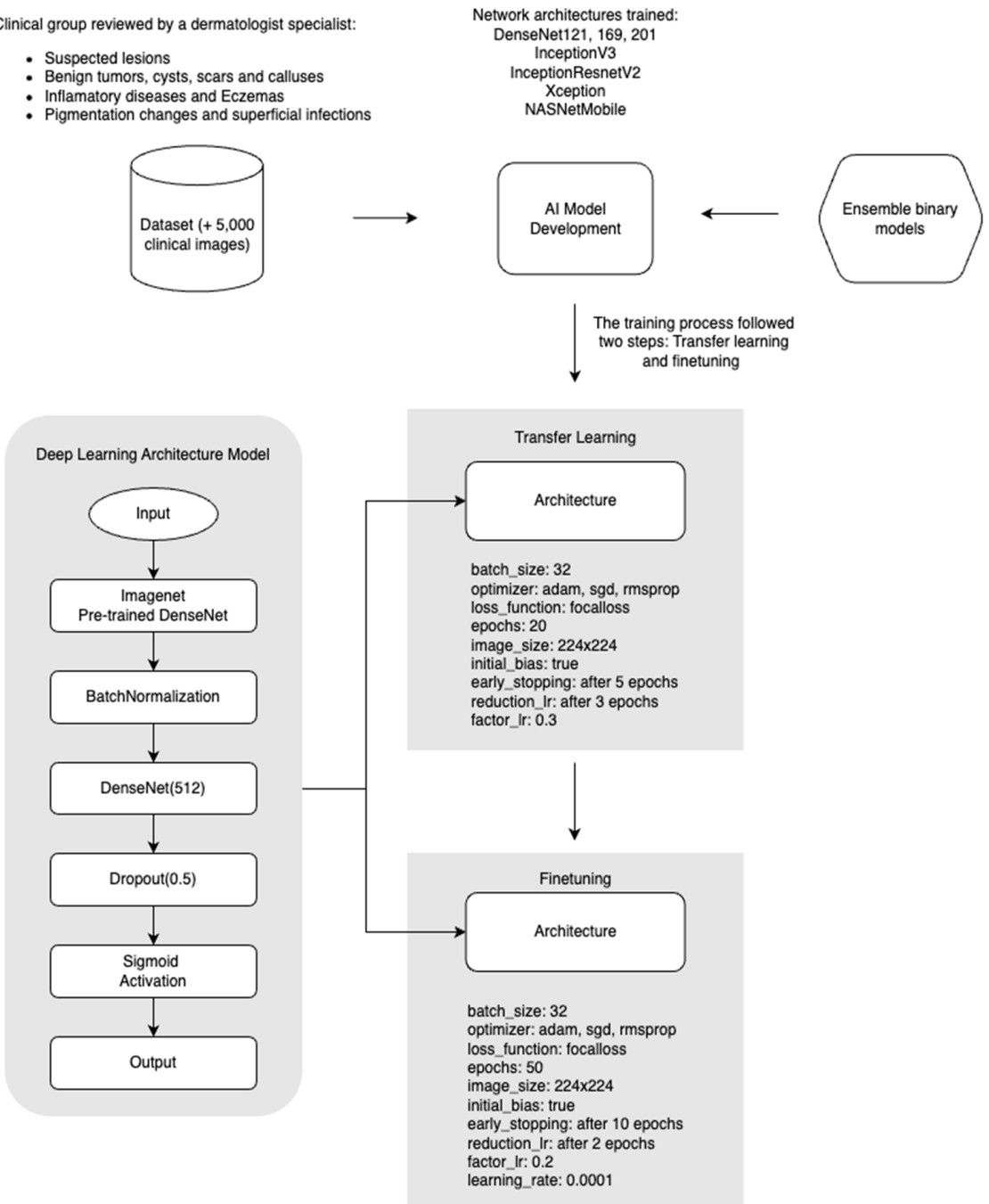

**Figure 4.** Workflow of the modeling process for classification of the disease groups.

### 3. Experimental Results

All models were trained on the Amazon AWS cloud platform, using a g4dn.xlarge NVIDIA T4 Tensor machine with 4vCPUs and 16GiB RAM. The best-performing model in the multi-classification training stage was the InceptionResnetV2 architecture, with a maximum accuracy of 0.75, sensitivity above 0.70, and a ROC curve above 0.90 for all clusters. In Figure 5, we present the respective values of the confusion matrix and the accuracy and recall for each cluster.

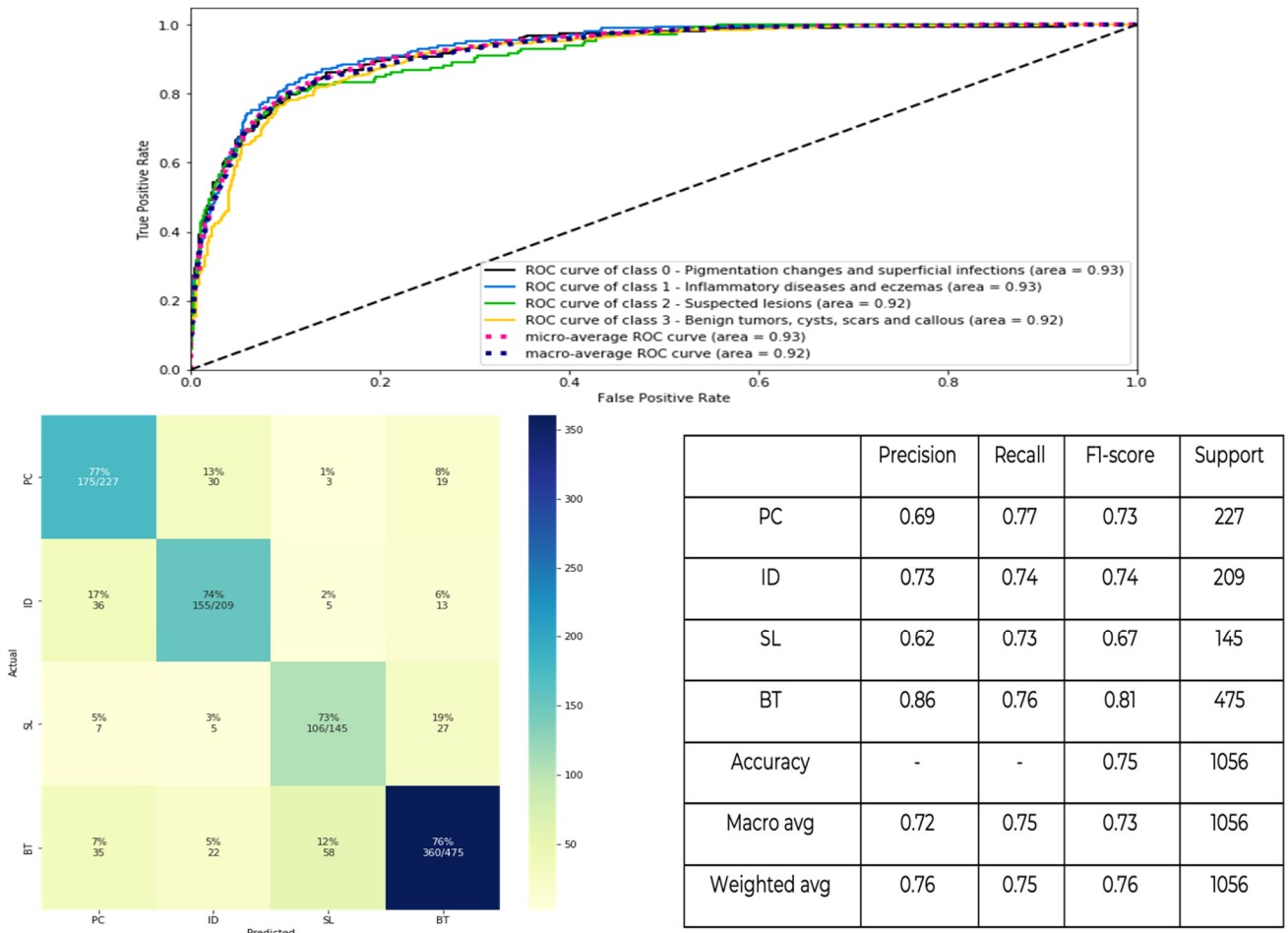

**Figure 5.** Results for multi-classification validation. In the figure, PC stands for pigmentation changes and superficial infections, ID for inflammatory diseases and eczemas, SL for suspicious lesions, and BT for benign tumors, cysts, scars, and calluses.

For the binary classification, the best performance was achieved by DenseNets models (Table 2).

**Table 2.** Results for binary classification validation; all AUC curves can be seen in Figure S1.

| Disease Cluster | Architecture | Accuracy | Recall/Sensibility | Specificity | Auc-Roc | Macro Precision | Macro F1-Score |
|---|---|---|---|---|---|---|---|
| Suspicious lesions | DenseNet169 | 0.82 | 0.88 | 0.81 | 0.91 | 0.70 | 0.73 |
| Inflammatory diseases and eczemas | DenseNet121 | 0.83 | 0.87 | 0.82 | 0.93 | 0.75 | 0.78 |
| Pigmentation changes and superficial infections | DenseNet169 | 0.81 | 0.87 | 0.79 | 0.92 | 0.75 | 0.77 |
| Benign tumors, cysts, scars, and calluses | DenseNet169 | 0.82 | 0.85 | 0.78 | 0.89 | 0.82 | 0.82 |

The creation of model ensembles in the literature has shown good results in competitions, including challenges in the identification of skin diseases and cancer, as demonstrated in [42]. With the results obtained from the binary models, we developed an ensemble of the best architectures for classifying our disease groups. Our prediction workflow prioritizes

suspicious (malignant) lesions when generating results from the ensemble model, with a threshold of 0.5. If the result falls below the threshold, the model will use the higher probability from the other models to make the final classification (Figure 6).

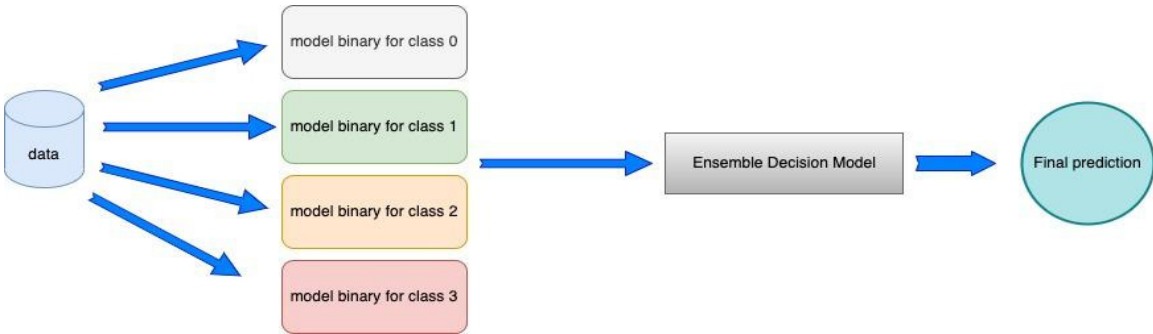

**Figure 6.** The ensemble model combines the base estimators for each binary decision and gives priority to suspicious lesions when estimating a final prediction.

The results presented in Figure 7 show the performance of the model when giving priority to suspicious lesions during prediction, while Figure 8 shows the performance of the model without this priority.

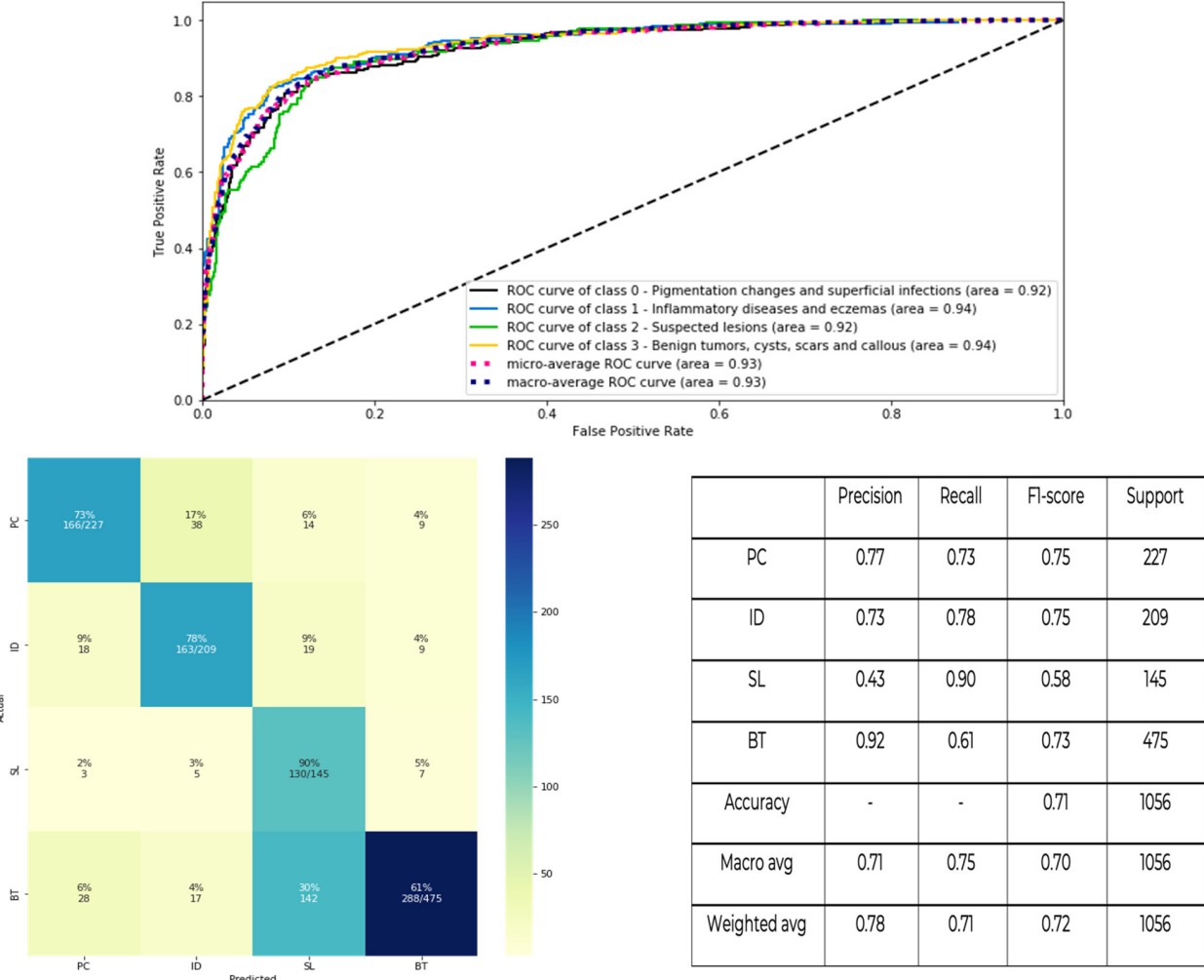

**Figure 7.** Priority for suspicious lesions.

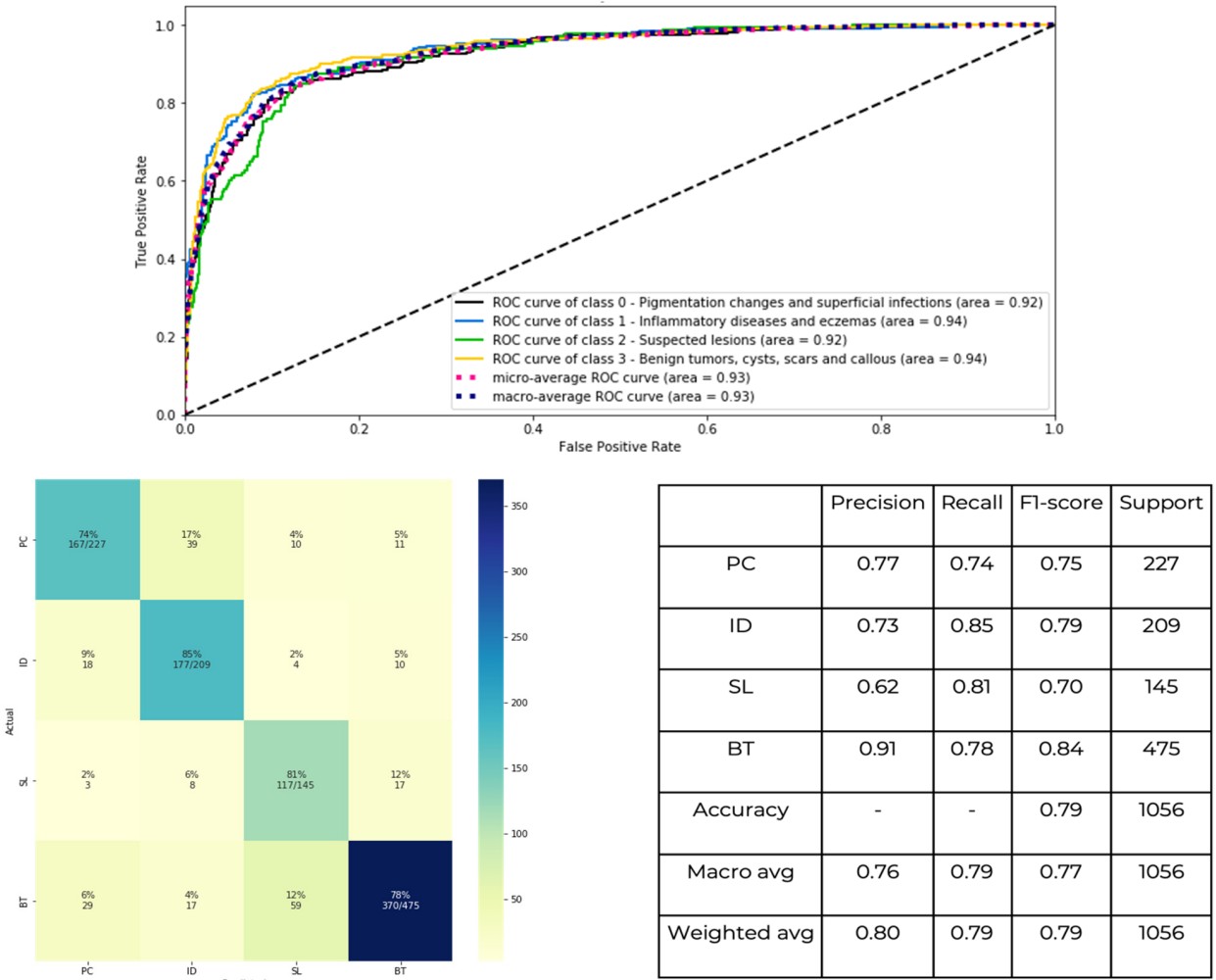

**Figure 8.** Equal triage for all group diseases.

Through visual explanations using the Grad-Cam algorithm, we were able to observe both the strengths and the weaknesses of the model. We were able to correctly analyze a lesion occupying more than one spot; moreover, any distinct object (for example, a ruler or clothing) that was close to the lesion did not seem to affect the result of the classification. However, dense hair areas, other skin injuries, parts of the body, and lesion illumination could affect the classification outcome (Figure 9).

We also attempted to validate the distribution of model predictions by gender and race both before and after training our model. Tables 3 and 4 show that there is no significant class bias present towards gender or any particular race.

**Table 3.** Distribution of model predictions separated by gender, both for the prediction and target variables. In the table, PC stands for pigmentation changes and superficial infections, ID for inflammatory diseases and eczemas, SL for suspicious lesions, and BT for benign tumors, cysts, scars, and calluses.

|  | Gender | BT | ID | PC | SL | Total |
|---|---|---|---|---|---|---|
| Predict | Male | 20.36% | 11.84% | 18.83% | 18.75% | 64.77% |
|  | Female | 9.85% | 8.43% | 9.00% | 7.95% | 35.23% |
| Target | Male | 31.16% | 10.98% | 12.97% | 9.66% | 64.77% |
|  | Female | 13.83% | 8.81% | 8.52% | 4.07% | 35.23% |

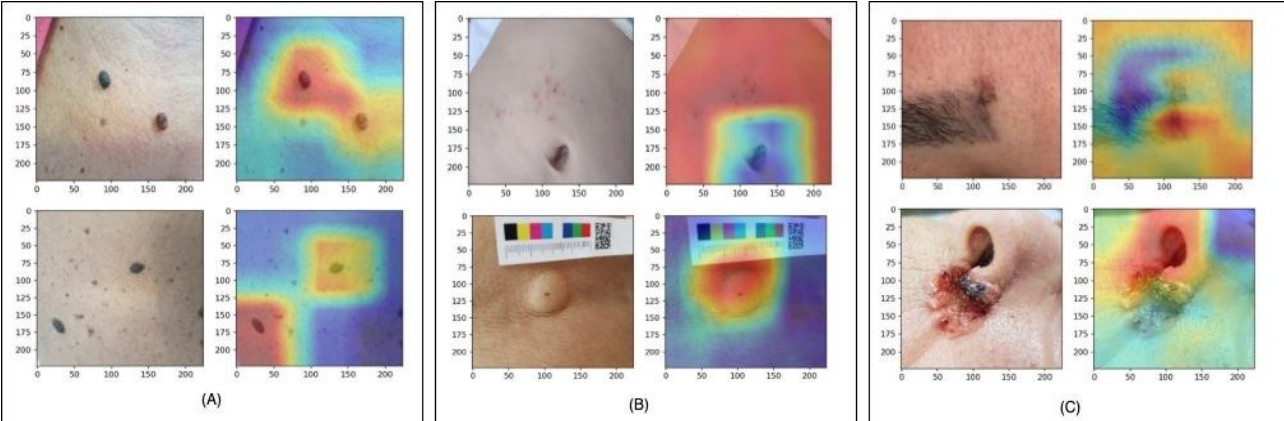

**Figure 9.** Visual explanation of the Grad-Cam algorithm. In (**A**), we observe how the model correctly classifies skin injuries occupying more than one spot. In (**B**), we observe that the model ignores objects that are distinct from the human body. In (**C**), we observe that the model has more difficulty dealing with hair and certain regions of the body such as the nose.

**Table 4.** Distribution of model predictions separated by gender, both for the prediction and target variables. The ethnic names are used according to IBGE (Instituto Brasileiro de Geografia e Estatística) nomenclature [43].

|  | Race | BT | ID | PC | SL | Total |
|---|---|---|---|---|---|---|
| **Predict** | yellow | 1.20% | 0.53% | 0.80% | 0.50% | 3.03% |
|  | white | 23.55% | 9.97% | 9.31% | 6.65% | 49.58% |
|  | n/a | 5.13% | 1.46% | 4.00% | 1.38% | 11.97% |
|  | brown | 12.82% | 8.81% | 8.52% | 4.07% | 30.20% |
|  | black | 2.12% | 0.64% | 1.72% | 0.52% | 5.22% |
| **Target** | yellow | 0.84% | 0.75% | 0.66% | 0.78% | 3.03% |
|  | white | 16.14% | 9.78% | 9.21% | 14.34% | 49.58% |
|  | n/a | 3.23% | 2.67% | 3.13% | 2.94% | 11.97% |
|  | brown | 8.74% | 6.27% | 7.69% | 7.50% | 30.20% |
|  | black | 1.68% | 1.13% | 1.94% | 0.47% | 5.22% |

## 4. Discussion

Before using our AI-powered application, patients are obliged to read and accept the terms of use. Those terms explain that the application is not 100% accurate and it cannot replace a physician's evaluation. Diagnosis can only be confirmed by a physician. We recommend that patients seek medical attention if they have doubts about the results or if they experience fever, pain, or intense itching, or if they are undergoing oncology treatment or have other chronic diseases.

We also explain that lesions covered by hair or tattoo pigmentation, or those located on the nails, in the genital area, inside the mouth, and on other body parts that are difficult to visualize, may be misclassified.

After analyzing the skin image using the artificial intelligence model, the application delivers the result by naming the predicted class with a color code: green, yellow, or red, along with a brief orientation on the suggested conduct for the patient. The GREEN color is related to the cluster of "Benign lesions, cysts, scars and callous", indicating that the person should follow up with general medical attention. The YELLOW color is related to "pigmentation changes and superficial infections" and "Inflammatory diseases and eczemas", indicating that the person should seek assistance from a dermatologist,

but should not consider the case as urgent. The RED color is related to the cluster of "Suspected lesions", indicating that the patient should prioritize assistance with a specialist dermatologist. The app will also permit multiple evaluations of the same lesion over a longer period of time.

The usage of "extra-equipment's" for obtaining skin images to analyze skin conditions represents the next level in achieving more promising results. As an example, we have [44], who used an otoscope to classify 11 diagnostic classes. They obtained a sensitivity of 99% on their test set, using deep-learning techniques.

Besides equipment and extra information, other associated data also play an important role in the field of development. High-frequency ultrasound was used by [45] to retrieve internal skin information that was used in a multimodal fusion network combined with clinical close-up images.

In order to avoid the risk of some details that are not part of the skin lesion being used as a factor for classifying lesions, complementary techniques, such as the prior segmentation of lesions, can also be applied, as demonstrated by [46]. Doing so, they obtained an accuracy of 87%.

## 5. Conclusions

In this project, we developed an AI model capable of analyzing skin images captured by mobile phone cameras. Our developed models achieved an overall accuracy of 0.79, sensitivities above 0.80, and a ROC curve of 0.90. We also developed a mobile phone application with the following functionalities: user registration, the capturing of skin images, real-time lesion classification, and reporting of the result of the classification with suggested clinical orientation.

The project's ultimate goal is to have a positive impact on the population by facilitating the early diagnosis and treatment of suspicious skin lesions, as well as guiding patients on the type of medical care they should seek regarding their skin diseases. By reducing waiting times for benign cases and prioritizing those with suspicious lesions, the project aims to optimize the population's skin health in the long term.

## 6. Limitations

- The AI model does not diagnose specific diseases; instead, it classifies the lesion into one of the specific groups.
- The model does not distinguish whether the image is a skin lesion or not; it assumes that a photo of the lesioned skin is presented.
- Depending on the distance from the lesion in the image, lesions may not be detected or can be misclassified.
- The model is not designed to estimate the size of the lesion in the image.
- Lesions in areas such as the eyes, nose, mouth, ear, navel, nipples, and genitals can make classification difficult, or these areas may be mistakenly confused with the lesion. Raised lesions may go undetected, and background objects may be confused with the lesion.
- Poorly lit, obstructed lesions, or blurred images of the lesions may not be classified correctly.
- The model developed is only capable of dealing with images (some additional metadata may enhance the classification).

**Supplementary Materials:** The following supporting information can be downloaded at: https://www.mdpi.com/article/10.3390/dermato4030011/s1. Table S1. Most relevant ICDs corresponding to each disease cluster. Figure S1. AUC-ROC Curves for all binary models.

**Author Contributions:** Conceptualization, A.P.d.S.; methodology, R.M.d.S. and A.L.O. and L.J.M.T.; validation, L.E.P.B.; software, E.J.R.-Z.; writing—review and editing, K.L.O. All authors have read and agreed to the published version of the manuscript.

**Funding:** this research received no external funding.

**Data Availability Statement:** No new data was created or analyzed in this study.

**Conflicts of Interest:** The authors declare no conflicts of interest.

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
