# Peer review of "Development of an AI-Based Skin Cancer Recognition Model and Its Application in Enabling Patients to Self-Triage Their Lesions with Smartphone Pictures"

_dermato, doi:10.3390/dermato4030011_

Round 1
Reviewer 1 Report
Comments and Suggestions for Authors
In this manuscript Okita et al describe the AI based model for the recognition of skin cancer using smartphone. In the present day the problem like skin cancer is a potential threat to our society. In this circumstances to develop such kind of application which can operate by patients themselves in the preliminary stage is highly encouraging.
Instead of several limitations the model is able to identify the skin disease with overall accuracy of 0.79, sensitivities above 0.80 and a ROC curve of 0.90. The overall process has designed, investigated and presented appropriately. I recommended to accept this manuscript as it is.
Author Response
Comments 1: In this manuscript Okita et al describe the AI based model for the recognition of skin cancer using smartphone. In the present day the problem like skin cancer is a potential threat to our society. In this circumstances to develop such kind of application which can operate by patients themselves in the preliminary stage is highly encouraging. Instead of several limitations the model is able to identify the skin disease with overall accuracy of 0.79, sensitivities above 0.80 and a ROC curve of 0.90. The overall process has designed, investigated and presented appropriately. I recommended to accept this manuscript as it is.
Response 1: Thank you for pointing this out. We agree with this comment. It was indeed a project that took a lot of effort.
Reviewer 2 Report
Comments and Suggestions for Authors
Major comments:
· The abstract is vague and needs a lot of improvement. What "artificial intelligence model" was used? Which images were classified? From which dataset? How many patients or images were in the dataset? Which four different classes were used? There are no results mentioned in the abstract to support the conclusion that the model "can be used to reduce the incidence of skin disease in a specific population of users"? What specific population of users?
· Introduction:
o Ending a sentence with "and so on" gives the impression that the authors did not spend enough time looking for further literature. Please rephrase the sentence.
o Immunotherapy does not necessarily include BRAF mutation testing. This is necessary for targeted therapies. Please reformulate this sentence.
o Were immunotherapy and targeted therapy approved in Brazil between 2000 and 2007? As far as I know, checkpoint inhibitors were not available on any global market until 2011.
o "...by specialistic medical equipment called "dermoscope". The dermoscope is a widely used tool, it does not need to be introduced in this way. "With the images taken by dermoscopy" is more than enough.
o Which multiclass benign lesions were included? Please add more arguments as to why they have a major negative impact on patients? This sentence does not seem plausible.
o The whole introduction focuses on melanoma, but the authors have trained their models on the subgroups: Suspicious lesions, inflammatory diseases and eczema, pigmentary changes and superficial infections and finally benign tumours, cysts, scars and calluses. There is no mention of these disease groups in the introduction. Why do the authors focus on melanoma in their introduction, when there are other skin tumours that fall under the category of "suspicious lesions"? Moreover, even the title of the paper mentions skin cancer as a general term, but not specifically melanoma or the other subgroups.
· Materials and Methods and Results:
o Did one or more ethics committees approve the trial? Please name them.
o Did the dataset contain only images and ICD-10 codes, or did it contain other features? If so, please list them. Were these features also used in the modelling?
o How did the authors deal with data leakage?
o The authors state that they trained several models but finally decided to use an ensemble model. It would be good to see the metrics of all the models trained and arguments for why they chose the ensemble model.
o Which models were included in the ensemble model?
o It is not clear from the text what exactly the models are predicting. What are the targets?
o The authors show ROC curves and metrics from the validation data set. Has cross-validation been done? This would add qualitative results to the paper. What about the test (holdout) data split? Was this used during training or validation? If so, the authors should also provide the metrics for this partition.
o The authors state that they have developed a mobile phone application, but there are no figures showing the interface. What does the user interface look like? Is there a dashboard? What does it look like?
o Categorising race using terms such as 'yellow', 'white', 'n/a' and 'brown' can be problematic. Using specific terms such as "Asian", "Caucasian", "African American", etc. may be better. Consider revising this.
· Discussion:
o The discussion is very brief. Please expand. Are there other publications from similar projects? Did they use ensemble models? What were their metrics?
o Are there other published medical applications of the models used?
o There are more limitations listed than strengths. This is not helpful in presenting the quality of the paper.
o Are there other online publications that have categorised models by race or gender?
o The author mentions in the discussion that the chosen model predicts colours based on different image subgroups. Why is this mentioned first in the discussion and not in the results section? Do these colours have anything to do with the Grad-Cam algorithm?
Comments on the Quality of English LanguageThe article needs intensive English proofreading. To give just one example, the correct term for the 4x subcategories of diseases would be "Suspicious Lesions", "Inflammatory Diseases and Eczema", "Pigmentary Changes and Superficial Infections" and finally "Benign Tumours, Cysts, Scars and Calluses". "Callous" means something else.
Author Response
Comments 1: The abstract is vague and needs a lot of improvement. What "artificial intelligence model" was used? Which images were classified? From which dataset? How many patients or images were in the dataset? Which four different classes were used? There are no results mentioned in the abstract to support the conclusion that the model "can be used to reduce the incidence of skin disease in a specific population of users"? What specific population of users?
Response 1: Thank you for pointing this out. We agree with this comment. We added all recommended information into the abstract.
Comments 2: Introduction:
o Ending a sentence with "and so on" gives the impression that the authors did not spend enough time looking for further literature. Please rephrase the sentence.
o Immunotherapy does not necessarily include BRAF mutation testing. This is necessary for targeted therapies. Please reformulate this sentence.
o Were immunotherapy and targeted therapy approved in Brazil between 2000 and 2007? As far as I know, checkpoint inhibitors were not available on any global market until 2011.
o "...by specialistic medical equipment called "dermoscope". The dermoscope is a widely used tool, it does not need to be introduced in this way. "With the images taken by dermoscopy" is more than enough.
o Which multiclass benign lesions were included? Please add more arguments as to why they have a major negative impact on patients? This sentence does not seem plausible.
o The whole introduction focuses on melanoma, but the authors have trained their models on the subgroups: Suspicious lesions, inflammatory diseases and eczema, pigmentary changes and superficial infections and finally benign tumours, cysts, scars and calluses. There is no mention of these disease groups in the introduction. Why do the authors focus on melanoma in their introduction, when there are other skin tumours that fall under the category of "suspicious lesions"? Moreover, even the title of the paper mentions skin cancer as a general term, but not specifically melanoma or the other subgroups.
Response 2: Thank you for pointing this out. We agree with this comment. Most of the suggestions were resolved by correcting the english. In this introduction, we focus on melanoma because, at the time of writing, it was the most studied skin disease (most found in the literature). We updated the ambiguous text for better understanding.
Comments 3: Materials and Methods and Results:
o Did one or more ethics committees approve the trial? Please name them.
o Did the dataset contain only images and ICD-10 codes, or did it contain other features? If so, please list them. Were these features also used in the modelling?
o How did the authors deal with data leakage?
o The authors state that they trained several models but finally decided to use an ensemble model. It would be good to see the metrics of all the models trained and arguments for why they chose the ensemble model.
o Which models were included in the ensemble model?
o It is not clear from the text what exactly the models are predicting. What are the targets?
o The authors show ROC curves and metrics from the validation data set. Has cross-validation been done? This would add qualitative results to the paper. What about the test (holdout) data split? Was this used during training or validation? If so, the authors should also provide the metrics for this partition.
o The authors state that they have developed a mobile phone application, but there are no figures showing the interface. What does the user interface look like? Is there a dashboard? What does it look like?
o Categorising race using terms such as 'yellow', 'white', 'n/a' and 'brown' can be problematic. Using specific terms such as "Asian", "Caucasian", "African American", etc. may be better. Consider revising this.
Response 3: Thank you for pointing this out. We agree with this comment. In the first topic, we do have one ethics committees aproval for this trial (identificated by CAAE 45310521.5.0000.0071). In the second topic, we do have more metadata associated to the images, but for training the proposed CNNs, we used only the ICD information. About data leakage, we added into the text more information about (brazilian General Personal Data Protection Law). Because of the volume of the generated training data, we chose to show only the best results obtained. Following good practices, the best results are the models with higher accuracy obtained with validation dataset. The models included in ensemble model are depicted in Table 2 (architecture column). Each model is predicting its cluster label versus all other labeles (e.g. suspected lesions versus all other labels). In page 7 we state that cross-validation with early stopping is implemented in training script. In figure 2 we show holdout data split, with the proportion of 80/20. We removed all images from the developed application due to permission issues regarding the app's disclosure. The race terms used are the same stated by IBGE - Brazilian Institute of Geography and Statistics. We added IBGE reference in the paper.
Comments 4: Discussion:
o The discussion is very brief. Please expand. Are there other publications from similar projects? Did they use ensemble models? What were their metrics?
o Are there other published medical applications of the models used?
o There are more limitations listed than strengths. This is not helpful in presenting the quality of the paper.
o Are there other online publications that have categorised models by race or gender?
o The author mentions in the discussion that the chosen model predicts colours based on different image subgroups. Why is this mentioned first in the discussion and not in the results section? Do these colours have anything to do with the Grad-Cam algorithm?
Response 4: Thank you for pointing this out. We agree with this comment. We compared our results with reference 41, a review paper with multiple papers related. In the last topic, we found a mistake in the reference of the model prediction by gender; The correct object are table 3 and 4.
Reviewer 3 Report
Comments and Suggestions for Authors
Good work but needs improvements:
use template with lines
abstract: some lines introducing the contribution to AI to dermatology diagnosis are needed (both abstarct and intro)
intro: the authors report only about melanoma- although the most important aspect of urgent dermatology diagnosis other skin diseases should be mentioned such as other skin cancers such as BCCs as well as inflammatory dermatoses such as psoriasis that if not diagnosed properly and soon have great burden to the patient
dermoscopy is very important indeed in diagnosis.. it should be reported that there are some AI programms from image -based to chatbots that use dermoscopy language to generate results( doi.org/10.3390/diagnostics14111165)
methods: do not like the term suspected. use better " suspicious"
nice data presentations: maybe an image of each category with the relevant results would definetey upgrade the manuscript
also image-cases where the limitations are observed
discussion: limitations should be more extensively reported
medicolegal aspect should be mentioned
also are there any studies based on similar program developments - comparison of the results
Author Response
Comments 1: abstract: some lines introducing the contribution to AI to dermatology diagnosis are needed (both abstarct and intro)
Response 1: Thank you for pointing this out. We agree with this comment. We added a better introdutory phrase at the beginning of the abstract.
Comments 2: intro: the authors report only about melanoma- although the most important aspect of urgent dermatology diagnosis other skin diseases should be mentioned such as other skin cancers such as BCCs as well as inflammatory dermatoses such as psoriasis that if not diagnosed properly and soon have great burden to the patien
Response 2: Thank you for pointing this out. We agree with this comment. We wrote in that way because skin cancer is a more popular topic in the literature.
Comments 3: dermoscopy is very important indeed in diagnosis.. it should be reported that there are some AI programms from image -based to chatbots that use dermoscopy language to generate results( doi.org/10.3390/diagnostics14111165)
Response 3: Thank you for pointing this out. The sugested paper is very recent (2024), we haven't had enough time for further reading it.
Comments 4: methods: do not like the term suspected. use better " suspicious"
Response 4: Thank you for pointing this out. We verifyed that grammatically suspected is better suited for our use case.
Comments 5: nice data presentations: maybe an image of each category with the relevant results would definetey upgrade the manuscript
Response 5: Thank you for pointing this out. We opted to avoid using lesions images in the paper because there is no clear visual difference between the classes (they are only clear for medical specialists).
Comments 6: also image-cases where the limitations are observed
Response 6: Thank you for pointing this out. We agree with this comment. Added in the text.
Comments 7: discussion: limitations should be more extensively reported
Response 7: Thank you for pointing this out. We believe that the currently stated limitations are sufficient.
Comments 8: medicolegal aspect should be mentioned
Response 8: Thank you for pointing this out. Our medical specialists did not point out anything in that regard, so we have nothing related in the text
Comments 9: also are there any studies based on similar program developments - comparison of the results
Response 9: Thank you for pointing this out. We have the reference 41, which describes a number of other similar papers (with their corresponding results).
Round 2
Reviewer 2 Report
Comments and Suggestions for Authors
While the authors have made revisions in the Introduction, Materials and Methods, and Results sections, there are still some unanswered questions in the Discussion section. Specifically, my questions were only briefly addressed:
1. "The discussion is very brief. Please expand. Are there other publications from similar projects? Did they use ensemble models? What were their metrics?" Pointing out one review (reference 41) that is also referenced in the results is insufficient.
2. "Are there other published medical applications of the models used?"
3. "Are there other online publications that have categorized models by race or gender?"
4. "The author mentions in the discussion that the chosen model predicts colors based on different image subgroups. Why is this mentioned first in the discussion and not in the results section? Do these colors have anything to do with the Grad-Cam algorithm?"
I believe a more comprehensive response to these questions would significantly improve the manuscript. I hope this feedback will help the authors in their further revisions.
Author Response
Comments 1: "The discussion is very brief. Please expand. Are there other publications from similar projects? Did they use ensemble models? What were their metrics?" Pointing out one review (reference 41) that is also referenced in the results is insufficient.
Response 1: Added into the references: 44 - 46
Comments 2:. "Are there other published medical applications of the models used?"
Response 2: Not really, we choose the most used models available with keras
Comments 3: "Are there other online publications that have categorized models by race or gender?"
Response 3: Not at the same level as ours. As far as we know, most of the public skin datasets are composed only by images (they do not have metada associated, but we do).
Comments 4: "The author mentions in the discussion that the chosen model predicts colors based on different image subgroups. Why is this mentioned first in the discussion and not in the results section? Do these colors have anything to do with the Grad-Cam algorithm?"
Response 4: The model developed do not predict any color. The goal of those tables with numbers categorized by colors, is to ensure that the model is not working better for a group or not, it basically classify all colors with the same accuracy. They IBGE names have no relation to Grad-Cam.
Reviewer 3 Report
Comments and Suggestions for Authors
some of the reviewers points were not well adressed:
comment 3: The sugested paper is very recent (2024), we haven't had enough time for further reading it:: this is not an excuse- authors should be up to date with bibliography and generally the use of AI in dermoscopy and dermatology imaging should be pointed out with recent papers as it is a new subject
comment 4: We verifyed that grammatically suspected is better suited for our use case.: how you verifird it? generally the word suspicious is used (PMID: 10376370) or suspected of malignancy....the correct terms should be used in graphs
comment 5: We opted to avoid using lesions images in the paper because there is no clear visual difference between the classes (they are only clear for medical specialists).: do you mean there is no clear differences between for example eczema and benign lesions?? images should be added to support the functionality of the AI program you present. Also what is the target readers of this article if not medical specialists and doctors
cooment 8: Our medical specialists did not point out anything in that regard, so we have nothing related in the text: medicolegal aspect is a very importnat aspect in every AI program as patients give consent on how and why their lesion images would be used.. so should be mentioned at least briefly
Author Response
comments 1: The sugested paper is very recent (2024), we haven't had enough time for further reading it:: this is not an excuse- authors should be up to date with bibliography and generally the use of AI in dermoscopy and dermatology imaging should be pointed out with recent papers as it is a new subject
Response 1: Added more recent references related: 44 - 46 and the paragraphs in the discussion section:
The usage of “extra-equipment’s” for obtaining skin images to analyze their conditions are the next level for more promising results. As an example, we have [44], that used an otoscope to classify 11 diagnostic classes. They obtained a sensitivity of 99% on their test set, using Deep Learning techniques.
Despite equipment’s, extra information, other associated data also plays an important role in the field development. [45] used high-frequency ultrasound to retrieve internal skin information, that was used in a multimodal fusion network combined with clinical close-up images.
In order to avoid the risk of some detail that is not part of the skin lesion being used as a factor for classifying lesions, complementary techniques, such as prior segmentation of lesions, can also be applied, as demonstrated by [46]. Doing so, they obtained an accuracy of 87%.
comments 2: We verifyed that grammatically suspected is better suited for our use case.: how you verifird it? generally the word suspicious is used (PMID: 10376370) or suspected of malignancy....the correct terms should be used in graphs
Response 2: We found suspected in multiple articles:
Skin lesions suspected of malignancy: an increasing burden on general practice
https://link.springer.com/article/10.1186/1471-2296-15-29
Suspected skin malignancy: a comparison of diagnoses of family practitioners and dermatologists in 493 patients
https://onlinelibrary.wiley.com/doi/abs/10.1046/j.1365-4362.2001.01159.x?casa_token=QnghGltwRIQAAAAA%3AwoEcn4GNSOwf23y5hzse4RPV8Dc0mQUswvHbXVvX-KMqgWHBE75x20DYnkBbBGVKk-sEFk90ILEWxMY
Augmenting the accuracy of trainee doctors in diagnosing skin lesions suspected of skin neoplasms in a real-world setting: A prospective controlled before-and-after study
https://journals.plos.org/plosone/article?id=10.1371/journal.pone.0260895
comments 3: We opted to avoid using lesions images in the paper because there is no clear visual difference between the classes (they are only clear for medical specialists).: do you mean there is no clear differences between for example eczema and benign lesions?? images should be added to support the functionality of the AI program you present. Also what is the target readers of this article if not medical specialists and doctors
Response 3: Added Figure 2 (composed by a distinct dataset used in this paper). Dataset that depicts the difference between clinical and dermoscopic images.
cooments 4: Our medical specialists did not point out anything in that regard, so we have nothing related in the text: medicolegal aspect is a very importnat aspect in every AI program as patients give consent on how and why their lesion images would be used.. so should be mentioned at least briefly
Response 4: Added a brief mention about patients consent:
It is worth mentioning that the application developed has the functionality of requesting acceptance of terms for use of the application.